# Comparative Evolutionary Epidemiology of SARS-CoV-2 Delta and Omicron Variants in Kuwait

**DOI:** 10.3390/v16121872

**Published:** 2024-11-30

**Authors:** Moh A. Alkhamis, Abrar Hussain, Fayez Al-Therban

**Affiliations:** 1Department of Epidemiology and Biostatistics, College of Public Health, Health Sciences Centre, Kuwait University, P.O. Box 24923, Kuwait City 13110, Kuwait; h.abrar@ku.edu.kw; 2Department of Public Health, Ministry of Health, P.O. Box 24923, Kuwait City 13110, Kuwait; drfayez@altherban.org

**Keywords:** SARS-CoV-2, B.1.617.2, B.1.1.529, ancestral state reconstruction, phylodynamic models, phylogeography, Kuwait

## Abstract

Continuous surveillance is critical for early intervention against emerging novel SARS-CoV-2 variants. Therefore, we investigated and compared the variant-specific evolutionary epidemiology of all the Delta and Omicron sequences collected between 2021 and 2023 in Kuwait. We used Bayesian phylodynamic models to reconstruct, trace, and compare the two variants’ demographics, phylogeographic, and host characteristics in shaping their evolutionary epidemiology. The Omicron had a higher evolutionary rate than the Delta. Both variants underwent periods of sequential growth and decline in their effective population sizes, likely linked to intervention measures and environmental and host characteristics. We found that the Delta strains were frequently introduced into Kuwait from East Asian countries between late 2020 and early 2021, while those of the Omicron strains were most likely from Africa and North America between late 2021 and early 2022. For both variants, our analyses revealed significant transmission routes from patients aged between 20 and 50 years on one side and other age groups, refuting the notion that children are superspreaders for the disease. In contrast, we found that sex has no significant role in the evolutionary history of both variants. We uncovered deeper variant-specific epidemiological insights using phylodynamic models and highlighted the need to integrate such models into current and future genomic surveillance programs.

## 1. Introduction

The emergence of the novel severe acute respiratory syndrome coronavirus 2 (SARS-CoV-2) at the end of 2019 in Wuhan, China, caused the most significant pandemic in recent history [1]. Unlike its past two predecessors—acute respiratory distress syndrome (SARS) and Middle East Respiratory Syndrome (MERS) that emerged in 2002 and 2012, respectively [2,3]—the SARS-CoV-2 rapid global spread led to unprecedented implications and permanent changes to human populations worldwide. These implications affected public health, the economy, environment, education, and human psychology worldwide [4]. The pandemic was declared a global public health emergency by the World Health Organization (WHO) in March 2020, and as of April 2024, over 700 million cases and 7 million deaths were reported worldwide [5]. Subsequently, the pandemic uniquely witnessed an unparalleled generation of genomic data for a single pathogen, including over 16 million full-genome sequences [6]. This plethora of genomic data was not only used to understand virus biology but also served a critical role in formulating and implementing effective pharmaceutical and non-pharmaceutical interventions [7].

Furthermore, SARS-CoV-2 infections were characterized by remarkable epidemiological heterogeneities among different geographical regions, attributed to differences in population demographics, environments, healthcare systems, government intervention measures, the emergence of more contagious viral variants, and other host and pathogen-related factors [8]. Thus, untangling the complex evolutionary epidemiology of rapidly evolving pathogens is critical for designing effective and efficient risk-based genomic surveillance systems. Traditional phylogenetic approaches were the standard tools for tracing variants’ genetic relationships and their transmission on the local and global scales. However, phylodynamic models have been the most robust investigative tools that flexibly combine genetic and related epidemiological factors on the same time scale to provide fundamental insights into the evolutionary history of the virus during the pandemic [9]. Therefore, the continuous integration of phylodynamic methods into current genomic surveillance programs will aid in gaining deeper evolutionary insights into the drivers of the SARS-CoV-2 dynamics and the potential future emergence of more pathogenic variants.

The formation of new viral variants/strains results from rapid mutation and deletion events on the gene segment that encodes the spike (S) protein [10]. Moreover, the World Health Organization (WHO) grouped SARS-CoV-2 major variants into three categories, including Variants Under Monitoring (VUM), Variants of Concern (VoC), and Variants of Interest (VoI) [11]. The Delta variant (B.1.617.2, or 21A, 21I, and 21J), which emerged in October 2020, was the most distinct VoC, as it caused notable mortalities and hospitalizations due to its high pathogenicity [12]. This was followed by the emergence of the Omicron variant in November 2021 (B.1.1.529, or 21K, 21L, 21M, 22A, 22B, 22C, 22D, 22E), which swiftly evolved into eight distinct strains within approximately a year and a half [11]. Yet, unlike the Delta variant, the Omicron variant was characterized by a high morbidity rate and significantly fewer hospitalizations [13]. However, due to the rapid mutation rate of the Omicron variant, it was described by notable immune escape to vaccination, including booster doses [13]. Finally, the Eris variant (EG.5) that emerged in February 2023 was designated as VUM, but it attracted worldwide attention as it had the highest mutation rate compared to other variants [14].

The first cases of COVID-19 in Kuwait were diagnosed in travelers, who were placed under immediate institutional quarantine on 23 February 2020. Since then, over 600 thousand infections, reinfections, and over 2500 deaths have been reported as of 10 April 2024 [5]. The pandemic caused significant local spreading and clustering events that were distinctly heterogeneous across different population demographics [15]. At the same time, the government’s intensive measures, including pharmaceutical and non-pharmaceutical interventions, notably slowed epidemic growth at the beginning of the pandemic [15,16]. However, the subsequent waves, dominated by SARS-CoV-2’s VoC, were substantially resistant to most public health interventions. Yet, declines in mortalities and hospitalization were mainly observed when the State of Kuwait reached 50% vaccination coverage from the general population [5,16].

Identifying temporal and spatial dynamics of COVID-19 among different population demographics on local scales based only on observed cases without genomic analysis might not truly reflect the true epidemiology of the pandemic in the affected communities [15]. This represents a critical gap for Kuwait and its neighboring countries, mainly as they were active air traffic hubs between continents, which facilitated the introduction of emerging variants [17]. Yet, throughout the pandemic, SARS-CoV-2 genomic sequencing activities have been widely implemented by many countries as a surveillance tool. However, most countries like Kuwait used genomic surveillance and classical phylogenetic methods (e.g., maximum likelihood trees) to identify only locally circulating viral variants and their phylogenetic relationships with global strains [18]. In contrast, Countries that used Bayesian phylodynamic methods in their sequence tracing programs have revealed novel epidemiological findings at local, regional, and global scales [19,20,21]. For example, phylodynamic models were implemented as a robust tool for exploring the evolutionary epidemiology of circulating variants by tracing their spatiotemporal origins [22,23,24], quantifying transmission events [25,26] and evaluating the effects of public health control and prevention programs before and after their implementation [27,28] in real time. That said, phylodynamic models have not been fully utilized in Arab countries except in Saudi Arabia [29] and other non-Arab East Asian countries [30,31]. Despite that, their robustness has been recently demonstrated on a regional level by simultaneously comparing and tracing the evolutionary epidemiology of several SARS-CoV-2 variants across the Arabian Peninsula [16].

Therefore, the primary objective of our study is to evaluate the current genomic surveillance program implemented by public health officials in Kuwait and integrate our phylodynamic analytical pipeline as a routine decision support tool. Here, we selected the Delta and Omicron variants as examples due to their distinct epidemiological features throughout the pandemic. We unveiled and compared the evolutionary dynamics and geographical origins of the Delta and the Omicron variants using all available genomic surveillance data sequenced in Kuwait and related metadata. We also revealed novel comparative insights into how different epidemiological characteristics, such as age and sex, shape Delta and Omicron’s evolutionary history and dispersal in Kuwait. These findings may improve current genomic surveillance activities on local scales and subsequently guide control and prevention efforts against current and future SARS-CoV-2 variants.

## 2. Materials and Methods

### 2.1. Sequence Retrieval and Curation and Dataset Processing

We explored the GISAID.org gene repository database and retrieved only all whole-genome SARS-CoV-2 sequences (i.e., with equal or greater than 29 kb in length) with their metadata isolated from cases reported in Kuwait (n = 1204) between 25 May 2020, and 16 October 2022. Next, we used the Nextclade web-based application [32] to identify and extract the Delta and Omicron isolates. Also, we exclude sequences designated as ‘bad’ by the application from the dataset. Furthermore, 100% identical sequences were discarded, as they are less frequent in the datasets (i.e., less than 5%). Additionally, 100% identical sequences do not boost the temporal signal of the dataset (i.e., molecular clock), hinder proper convergence of the phylodynamic model’s posterior estimates, and significantly amplify the costs of required computational resources [33,34]. Hence, the final combined Delta and Omicron dataset comprised 726 sequences isolated between 24 May 2021, and 16 October 2022. We then downloaded the metadata of the genomic data used to reconstruct the Novel Coronavirus-Global Subsampling (n-cov) tree from the Nextstrain [35] web interface (https://nextstrain.org/ncov/gisaid/global/, accessed on 25 September 2023) to retrieve the corresponding sequences from GISAID.org website. At the time of data retrieval, the NextStrain genomic dataset comprised 3196 (excluding Kuwaiti isolates) representing the reference global sequences collected between 26 December 2019 and 11 October 2022. Therefore, the final combined dataset (i.e., Kuwait and the global) comprised 3922 sequences (see GISAID Appendix A for the accession numbers and Epi_set identifier). In this study, we refer to the sequences isolated in Kuwait as focal sequences, while the reference sequences isolated globally as context sequences, based on the terminology suggested by NextStrain authors [35]. Moreover, the NextStrain algorithm handles millions of published sequences worldwide by frequently subsampling predefined continental regions within every 1, 2, and 6 months while maintaining a group of earlier reference strains to generate a representative global tree that is computationally efficient to be visualized by the average user. Therefore, we deemed the NextStrain sequences as our context genomic dataset because they ensure a more equitable representation of global sequences sampled across different geographical regions and periods in parallel to the time frame of the selected focal sequences from Kuwait.

We used genome-sampler version 2.0 [36] to subsample the context dataset described above and reduce the computational costs of the subsequent analyses. The genome-sampler algorithm excluded 1069 sequences (=33.26%) while preserving the dataset’s representation in terms of the time and location of their isolation and overall genomic diversity. Thus, the resulting dataset comprised 2853, including context and focal sequences.

### 2.2. Preliminary Phylogenetic Analysis

We used MAFFT version 7.49 [37] to align the sequence dataset. We also used AliView version 1.74 to visualize the resulting alignment and trimmed the first 130 bp and the last 50 bp to reduce potential sequence artifacts, as suggested elsewhere [35]. Recombination Detection Program version 4.0 [38] was used to confirm the absence of recombination events in the alignment. We used IQ-tree version 2.0 [39] to construct the subsequent analysis’s maximum likelihood (ML) trees using 1000 bootstraps and 100 replicate runs. Additionally, using the model assessment procedure integrated into IQ-Tree, we found that the general time-reversible with empirical base frequencies and gamma-distributed varying rate model combination was the best-fitting substitution model for the whole selected sequence data. To further ensure that the selected context sequences by the genome-sampler algorithm were representative and accommodating of the genomic variation of the focal sequences, we compare the ML tree topologies before (n = 3922) and after (n = 2853), removing the surplus context sequences (Figure 1A).

We then extracted the clades corresponding to the Delta and Omicron variants, where most focal sequences are clustered and reconstructed an independent ML tree for each variant (Figure 1A). Tempest version 1.5.3 [40] was used to run the root-to-tip regression models to assess the presence of sufficiently strong temporal signals for both variants by estimating the correlation coefficient value (R^2^). We found that the phylogeny of both variants exhibited sufficient temporal signal (R^2^s > 0.3; Figure 1A) to allow the reconstruction of the subsequent molecular clock models. Further, we used TempEst to investigate and exclude sequence outliers and errors in the datasets. Moreover, we used TreeTime [41] to calibrate the temporal scale for the phylogeny of each variant. The two major reference genomes isolated in Wuhan in 2019 (i.e., hCoV-19/Wuhan/Hu-1/2019 and hCoV-19/Wuhan/WH01/2019) were used as an outgroup for inferring the posterior parameters of the subsequent phylodynamic models. The final Delta dataset comprised 496 sequences, while the Omicron dataset comprised 648 (see Appendix A for their detailed profile).

### 2.3. Inferring the Temporal Dynamics of Variants in Kuwait

We retrieved the SARS-CoV-2 observed case data in Kuwait from the WHO COVID-19 Dashboard webpage reported between January 2021 and November 2022 [5] to characterize the temporal dynamics of the variants within their dominating periods. This selected period encompasses the combined time of isolation of each variant dataset described above. This was achieved by constructing an epidemic curve from the observed data to quantify the time-dependent reproductive number (R_td_) model [42]. The method is implemented in R package ‘R0’ [43]. It is based on a likelihood procedure that estimates R_td_s for each point of time (i.e., daily R_td_s) throughout the epidemic by averaging all potential transmission networks associated with the observed cases. We inferred the generation time from the epidemic curve using a serial interval with a log-normal distribution, a mean of 4.7, and a standard deviation of 2.9 days [44]. Additionally, we used 10,000 simulations to obtain the 95% confidence interval (CI) for each inferred daily R_dt_.

### 2.4. Reconstructing Variants’ Demographic History

We modelled the demographic history and inferred divergence times of the Delta and Omicron variants in Kuwait using the Bayesian relaxed-molecular clock models implemented in BEAST software package version 1.10.4 [45]. Besides, the BEAGLE library [46] was used to expedite the computational process of our phylodynamic models. We used the substitution model selected by IQ-tree and assessed commonly used three parametric and one non-parametric node-age priors (i.e., tree models). The parametric priors included the constant population size (CP) [47], exponential (EG) [48], and expansion growth (ExG) [48], while the non-parametric was the Bayesian Skygrid (SG) [49]. In addition, we assessed two branch-rate priors for each node-age prior model, including the uncorrelated lognormal (UCLN) and exponential (UCED) distributions [50]. The continuous-time Markov chain (CTMC) hyperprior [51] integrated into BEAST was used to compute the posterior parameters of the selected priors’ distributions. Thus, we compared eight candidate prior combinations to choose the best-fitting model for each variant. To conduct the Bayes factor (BF) comparison for the best-fitting model, we used the estimates calculated from the marginal likelihood distributions by the path-sampling (PS) and stepping-stone (SS) procedures [52]. Here, we found that the UCED+ExG priors were the best-fitting model for both the Delta and Omicron variants (BFs > 2) and, therefore, were used for the subsequent ancestral trait reconstruction analyses (see Appendix A).

However, we used the UCED+SG model to infer each variant’s effective population size over time. We then compared the patterns of effective population sizes between all sequences combined (i.e., context and focal sequences) and a selected monophyletic clade circulating predominantly in Kuwait for each variant. For the selected monophyletic clades from Kuwait, we ran an independent UCED+SG model analysis.

### 2.5. Ancestral Trait Reconstruction for Phylogeography and Transmission Between Hosts

We inferred the global origins of each viral variant. We identified its significant transmission routes between Kuwait and the other six continents, including Europe, Asia, Africa, Oceania, and North and South America, using the ancestral trait analysis extension implemented in BEAST [53]. Moreover, we identified transmission routes and their directionality between the geographical locations using the Bayesian stochastic search variable selection (BSSVS) procedure [54]. Additionally, we assessed the fit sequence data to the symmetric (reversible transitions) and asymmetric (irreversible transitions) models using the BF comparisons described above. Further, we investigated the intensity of viral jumps between geographic traits by computing the expected number of forward and backward exchanges using the Markov-jump (MJ) approach [55]. This analysis was also repeated to host characteristics to estimate the transmission cycle between three different age groups (including those under 20, between 20 and 50, and above 50 years old) and sex (including males and females) in Kuwait. Thus, we interrogated three discrete traits, including geographic, age, and sex origins for the isolates of each variant.

We inferred the phylodynamic models’ posterior parameters using the Bayesian Markov chain Monte Carlo (MCMC) simulations for 300 million cycles and sampling every 30000th state. Tracer version 1.6 [56] was used to evaluate the effective sample size (ESS) to assess the proper convergence (i.e., ESS > 200) of each MCMC chain. Further, we conducted duplicate MCMC runs for each candidate model to evaluate the consistency of the marginal likelihood estimates. From each MCMC chain, 10% of the sampled parameters were discarded as burn-in, and the resulting posterior median node heights (i.e., their posterior probability density) were presented as the maximum clade credible (MCC) tree generated by TreeAnotator. Tracer was also used to generate the Skygrid plots for each variant and its selected subsequent local monophyletic clades described above. Besides, we used SPREAD3 version 0.9.6 [57] to interrogate the significance of the non-zero rates of viral dispersal routes inferred by the BSSVS procedure. We then plotted viral dispersal with BSSVS BF greater than three for each discrete trait. Finally, Bayesian Tip-Significance Testing (BaTS) version 2.0 [58] was used to calculate the Parsimony scores (Ps) and Association indices (Ai) and their *p*-values from a sample of 2000 posterior trees and 200 null replicates to assess the role of the selected discrete traits in shaping the structure of the inferred MCC tree for each variant.

## 3. Results

### 3.1. Demographic History and Temporal Dynamics

Our results indicate that the Omicron variant had a remarkably higher mean nucleotide substitution rate per site per year (1.40 × 10^−3^, 95% HPD [1.24 × 10^−3^, 1.57 × 10^−3^]) than the Delta variant (7.57 × 10^−4^, 95% HPD [6.73 × 10^−4^, 8.34 × 10^−4^]). Besides, our Skygrid analyses revealed sequential peaking in the effective population sizes through time for both variants (Figure 2A,C). Moreover, many inferred genetic diversity peaks mirrored most of the significant transmission events estimated by the time-dependent reproductive number model (R_td_s > 1; Figure 1B and Figure 2A,C), particularly for the Omicron variant. In fact, our comparative results of mixed global and local demographic reconstructions for the genetic diversity indicate consistent, steady peaking for the delta variant followed by a stabilized effective population size (Figure 2A), while erratic, inconsistent episodes of peaking with no sign of decline or stabilization was observed in the effective population size of the Omicron variant (Figure 2C) throughout the study period. This remarkably mirrors the pattern of the inferred R_td_, where a small number of significant transmission events were inferred during the period of Delta dominancy (R_td_s ≥ 1), while erratically large super-spreading events were inferred during the period of Omicron dominancy (R_tds_ ≥ 2; Figure 1B). Similarly, the demographic reconstruction of the selected local Delta clade showed a steady consistent increase followed by a stabilization period in the population size (Figure 2B). This demographic pattern corresponds to no significant spreading events inferred for the observed reported cases (R_td_s ≤ 1; Figure 1B). In contrast, the selected local Omicron clade had a predominant period of increase in the population size (Figure 2D) that corresponded to super-transmission events in the community (R_td_s ≥ 2; Figure 1B).

### 3.2. Phylogeographic History

Our BF comparisons suggested that the asymmetric phylogeographic model with irreversible transitions was the best-fitting discrete trait model (BFs > 10) for both variants. Additionally, the Ps and Ai estimates and their significant *p*-values (<0.01) of the selected discrete geographic traits were statistically significant in shaping the structure of the inferred phylodynamic tree for each variant (Table 1). We inferred that the introductions of the Delta variant in Kuwait originated from East Asia with strong posterior node support and root-state posterior probability (RSSP = 96) during January 2021 (95% HPD [November 2020, February 2021]; Figure 2A). Since the Delta was introduced to Kuwait, our Bayesian Skygrid analysis of the locally dominant circulating clade showed plateauing effective population size between April and July 2021, followed by a steady stabilization period throughout the year until early 2022 (Figure 2B). However, results also showed that the introductions of the Omicron variant originated from North America (RSPP = 0.89) in August 2021 (95% HPD [May 2021, November 2021]; Figure 2C). Our demographic reconstructions of the combined global and local sequences and the selected clade dominant in Kuwait post-introduction showed continuous inclines in the population size with no apparent signs of stabilization or decline until October 2022 (Figure 2C,D).

The BFs inferred by the BSSVS procedure revealed that the most significant unidirectional geographical dispersal route for the Delta variant was from East Asia to Kuwait (BF > 100; Figure 2B), whereas the least significant unidirectional route was inferred from Oceania to Kuwait (BF < 100; Figure 2B). Also, East Asia had the highest mean counts of relative forward transitions into Kuwait (forward = 24 vs. reverse = 4; Figure 2E). Other inferred dispersal routes from the remaining geographical regions for the Delta variant had a BF < 3 and almost zero intensity of viral jumping between regions (MJs < 1). In contrast, the most significant unidirectional dispersal route for the Omicron variant was inferred from Kuwait into East Asia (BF > 100; Figure 2D), which also had the highest relative reverse transitions (reverse = 23; Figure 2F). The least significant unidirectional route was inferred from Africa to Kuwait (BF < 100; Figure 2D), with relatively lower intensity of forward and reverse transitions between regions (MJs ≤ 5). Other mid-significant dispersal routes (BFs between 100 and 10) with relatively lower intensity of viral jumping (MJs ≤ 7) were estimated from Kuwait to Europe and from North America to Kuwait (Figure 2D).

### 3.3. Transmission Between Hosts

We found strong evidence that our selected age groups had a critical role in shaping the transmission and phylogeny of both variants among infected patients. Moreover, the BF comparisons indicated that the asymmetric discrete trait model was the best-fitting model for the sequence data of both variants (BFs > 12). In addition, the Ps and Ai inferences of the age groups and their significant *p*-values (<0.05; Table 1) suggest that they did contribute to the structure of the posterior MCC trees (Figure 3A,B). For both variants, adult patients aged between 20 and 50 years old were remarkably dominant as the most likely ancestral host for transmitting the viruses to other age groups (RSSPs > 0.8; Figure 3A,B). At the same time, the BSSVS BF revealed that adults were the central host for the dispersal routes of variants to other age groups (BSSVS BF > 3; Figure 3A,B). These significant transmission routes were notably intense, with inferred relative forward transitions between adults on one side and other age groups on the other side (MJs ≥ 85). In contrast, the least significant and intense dispersal route was inferred from seniors (i.e., age > 50) to children and adolescents (i.e., age < 20) for the Delta variant (Figure 3A). Nevertheless, we found a significant and relatively intense dispersal route from seniors to adults for the Omicron variant (BSSVS > 100; MJs = 203). However, no significant or intense transmission routes were inferred from children and adolescents to other age groups for both variants (Figure 3A,B).

Finally, our BF comparison illustrated that the symmetric discrete-trait model with the reversible transition between the sexes was the best-fitting model for both variants’ datasets (BF > 5). Yet, the Ps and Ai estimates and their remarkably insignificant *p*-values (>0.1; Table 1) demonstrated that sex had no important role in shaping the structure of the inferred MCC tree (Figure 3C,D). In fact, while the ancestral viruses were isolated from males (Figure 3C,D), the transmission routes between the two sexes were almost equally significant (BSSVS > 1000) and intense (MJs ≥ 348) in later time periods.

## 4. Discussion

In this study, we used a rigorous Bayesian phylodynamic modelling approach to interrogate and compare the evolutionary epidemiology of the Delta and Omicron variants in Kuwait. Parallel to the epidemic progression in the community, we revealed critical insights into the demographic history of each variant and identified geographical origins and significant migration routes between Kuwait and other continents. Moreover, we untangled new insights into the transmission dynamics of the variants by quantifying the evolutionary role of age and sex in shaping their dispersal routes between hosts. Yet, these findings reflect a mixture of both variants’ global and local evolutionary characteristics with particular emphasis on viruses isolated in Kuwait. Our results are essential for guiding current and future SARS-CoV-2 genomic surveillance programs on local and regional scales and encourage continuous use and integration of phylodynamic methods into public health decisions on such notorious, rapidly evolving pathogens.

We inferred that the Omicron variant remarkably surpassed the Delta variant in terms of evolutionary rate, population growth, and transmission magnitudes on global and local scales (Figure 1B and Figure 2). These findings support the notion that each emerging VoC had relatively higher evolutionary advantages than its predecessor (or ancestral) variants and subvariants [7,59]. Moreover, our discrete phylogeographic model indicated that the Delta and the Omicron variants’ epidemic waves in Kuwait resulted from multiple direct introductions from East Asia and Africa, respectively, where the viruses were first identified (Figure 2) [60,61]. These results also agree with past observations on emerging VoCs’ strong genetic structure, which is mainly maintained by their geographic and temporal origins [16,62]. For the Delta variant, we found a notably significant and intense (BF-BSSVS > 100, and Forward MJ = 25; Figure 2B,E) unidirectional migration route from East Asian countries into Kuwait. In contrast, the statistically unsupported (BF-BSSVS < 10) and least intense (MJ = 4) migration route was inferred from Kuwait to East Asia (Figure 2B,E). This is not surprising since Kuwait’s local population comprises a striking proportion (≈69%) of migrant workers flowing continuously from India, Bangladesh, the Philippines, and Pakistan into Kuwait and vice versa [15]. In fact, Kuwait’s public health authorities imposed drastic air traffic restrictions on people flowing from such countries between March 2020 and February 2021 [63]. However, this restriction was then eased gradually afterwards, which coincided with the multiple introductions of the Delta variant in Kuwait, as demonstrated in Figure 2A. Air traffic restrictions were entirely lifted in October 2021, particularly for the vaccinated, in which we inferred significant migration routes that mirror the regular travel routes for the Kuwait population to Europe, North America, Africa, and vice versa. It is worth noting that while Kuwait is a relatively small country, it constitutes a vital travel hub for migrant workers and exports of Oil in the Middle East [64].

Our results for the Delta variant showed remarkable stabilization in the number of daily cases and insignificant spreading events in Kuwait (R_td_S ≈ 1; Figure 1B) that might be attributed to the large vaccination rollout at the beginning of 2021. In fact, Kuwait has a very distinguished high vaccination record during the pandemic, in which the country has administrated over 8 million doses throughout 2021 and 2022, which approximates a vaccination coverage of 96.5% of the total population [65]. However, we inferred that Kuwait’s vaccination resources were ineffective against the Omicron variant, in which results illustrate an unprecedented number of infections and multiple significant super spreading events (R_td_S > 1; Figure 1B). Indeed, the Omicron variant exhibited a significantly higher evolutionary rate than the Delta variant and went through a momentous antigenic shift that facilitated its vaccine escape [66]. Therefore, the significant drop in the observed cases and super spreading events were mainly attributed to the resulting natural immunity from infection and reinfection, particularly for the Omicron variant [67].

One notable finding inferred from the analysis of the temporal dynamics of the observed cases in Kuwait for both variants was that their epidemic curve and active transmission events were characterized by two distinct peaks (approximately around March and July of the years 2021 and 2022, respectively; Figure 1B) mirroring the results of a past phylodynamic study in the Arabian Peninsula [16]. These peaks coincided with seasonal events such as sandstorms and other wind-related climate events [68], frequently occurring between mid-March and early August every year. Therefore, it was inferred that air pollution and related anthropological activities shaped our observed epidemic curves for both variants [16]. Additionally, our analysis revealed a remarkable similarity in the periods and patterns of population growth through time for locally selected clades coinciding with the climatic events in Kuwait. For both variants, we showed that the relative genetic diversity, through time, starts between March and April and undergoes a steady prolonged growth that peaks and stabilizes in July (Figure 2B,D). Thus, like other viral respiratory infections, air pollution and seasonality should be considered in future phylodynamic studies that implement generalized linear models to account for such critical evolutionary predictors of newly emerging variants [69].

One unique aspect of the present study, particularly on the level of the Arabian Peninsula and the Middle East, is the direct incorporation of some important host characteristics (also known as the most common confounding variables), such as sex and age, into the statistical framework of our phylodynamic model. We demonstrated that sex has no significant role in shaping the evolutionary epidemiology of the Delta and Omicron variants (Table 1) and that equally intense and significant transmission pathways for the virus were inferred between males and females (Figure 3C,D). These findings add further insights to the results of past studies that attempted to assess the role of sex on the transmission and infection of SARS-CoV-2, which either found that sex was statistically insignificant or was primarily inconclusive about its biological role [70,71,72]. Nevertheless, age has a remarkably significant role (Table 1) in shaping both variants’ evolution and transmission (Figure 3A,B). These results illustrate that adults between 20 and 50 years are focal in the transmission cycle to other age groups. In contrast, children and adolescents under 20 years old were consistently a destination for the virus dispersal from adults (Figure 3A,B). In terms of biological plausibility, our results agree with the notion that children express substantially lower angiotensin-converting enzyme 2 (ACE-2) receptors in their lungs than adults, rendering them less susceptible to infection with notably milder symptoms upon infection [73]. Thus, like past studies, we conclude that children were not super-spreaders of either Delta or Omicron variants. However, in Kuwait, the public health authority closed schools for in-person learning for approximately 18 months (i.e., between February 2020 and October 2021) from 1st to 12th grades [63], based on the assumption that children are important super-spreaders for COVID-19 infections. Therefore, utilizing our robust analytical approach to inform and guide such impactful decisions for future pandemics is critical to avoid their short and long-term implications on such vulnerable populations [74].

In the present study, we used only related sequences available at GISAID within the study’s time frame that corresponds to the emergence of the Delta and Omicron variants. This is because all sequences isolated from Kuwait were mainly published in GISAID, and we found only a few isolates published in the GenBank that were similar to our retrieved dataset. This limitation may restrict and bias our results for the genetic dynamics of the variants in space and time. However, this is an inherent limitation of most phylogenetic studies, but using a Bayesian statistical framework allowed us to accommodate the uncertainties around our inferences, which is a remarkable advantage of phylodynamic methods. Furthermore, we could not retrieve the metadata for the travel history of sequenced patients as they were scarce in the selected local dataset. Therefore, this limitation prevented us from using the BEAST software extension created specifically to accommodate individual travel history to generate more realistic and less biased posterior results for SARS-CoV-2 migration history across countries [75]. Lastly, one would argue that our inferences related to the transmission history of both variants between different age groups were attributed to the lower number of sequences for a given age group (e.g., for the Delta variant, patients aged 50–20 years old had 249 isolates, while patient aged <20 had 78; see Appendix A). Nevertheless, the results of the phylogeographic analysis demonstrated that our analyses were utterly insensitive to the number of sequences obtained by the selected discrete traits. For the example of the Omicron variant, most of the included isolates were from Kuwait (n = 383; Appendix A), but the inferences of ancestral trait reconstruction inferred that the origins of the viral introductions were either from Africa (n = 35) or North America (n = 80; Figure 2C and Appendix A). This is in addition to the biological plausibility of our inferred findings related to age, described above.

To this date, public health authorities in Kuwait and most neighboring countries in the region continue to rely on traditional genomic surveillance of SARS-CoV-2 in both the pandemic and endemic stages. Additionally, their phylogenetic analysis methods mainly aim to identify whether a new variant was introduced to the country and its prevalence compared to earlier variants. This approach lacks critical elements for intervention decision support, leading to both long and short-term public health and economic implications. The rigorous implementation of our present phylodynamic analytical pipeline provides a rich landscape for understanding the complex epidemiology of rapidly evolving pathogens and, subsequently, designing an effective and efficient risk-based genomic surveillance system. Here, using a comparative phylodynamic approach allows for the proper allocation of intervention resources upon the emergence of new variants. For example, not all emerging variants exhibit highly pathogenic properties like the Delta variant. At the same time, not all variants have intense evolutionary characteristics in terms of mutation rate and transmission like the Omicron variant. Hence, the magnitude of the intervention should mainly rely on the evolutionary characteristics of the emerging variant. Finally, it is important to note that the evolutionary parameters (e.g., branch-rate or node-age prior models) of rapidly evolving viral pathogens might change through time, especially when sampling is done over shorter durations with the introduction of new isolates to the analysis, potentially leading to the selection of different best-fitting phylodynamic models and, consequently, posterior inferences [76]. Thus, revisiting the present analytical pipeline is critical for sustaining future public health intervention efforts guided by genomic surveillance.

## 5. Conclusions

This study represents the first attempt to implement variant-specific comparative phylodynamic models within local SARS-CoV-2 genomic surveillance efforts in Kuwait. We untangled and compared the demographic and phylogeographic history of the Delta and the Omicron variants under both public health restricting and easing stages on local and global scales. Our phylogeographic analyses indicated that the Delta and the Omicron variants’ epidemic waves in Kuwait resulted from multiple direct introductions from East Asia and Africa, respectively, where the viruses were first identified. Additionally, Delta’s dispersal rates from Eastern countries were substantially significant and intense, mirroring the structure of the Kuwait population, which comprises approximately 69% of migrant workers with their respective air traffic movements from their country of origin. However, the Omicron variant did not only replace the Delta’s viral population but also exhibited a notably higher evolutionary rate. We found that climatic and anthropological seasonal activities have coincided with the sequential patterns of genetic diversity through time of both variants in Kuwait.

Most importantly, we showed that sex has no role in shaping the evolutionary history of both variants. However, adults aged between 20 and 50 were focal in spreading the virus to older and younger age groups. Thus, we were able to refute the notion that children and adolescents were superspreaders of the virus to other age groups. Because SARS-CoV-2 is considered the fastest-evolving pathogen observed in human history, continuing risk-based genomic surveillance efforts with the integration of phylodynamic models is critical for efficient and effective guidance of public health decision-making. However, increasing high-quality sequencing capacity and detailed reporting of their related metadata is essential for successfully implementing such intervention efforts.

## Figures and Tables

**Figure 1 viruses-16-01872-f001:**
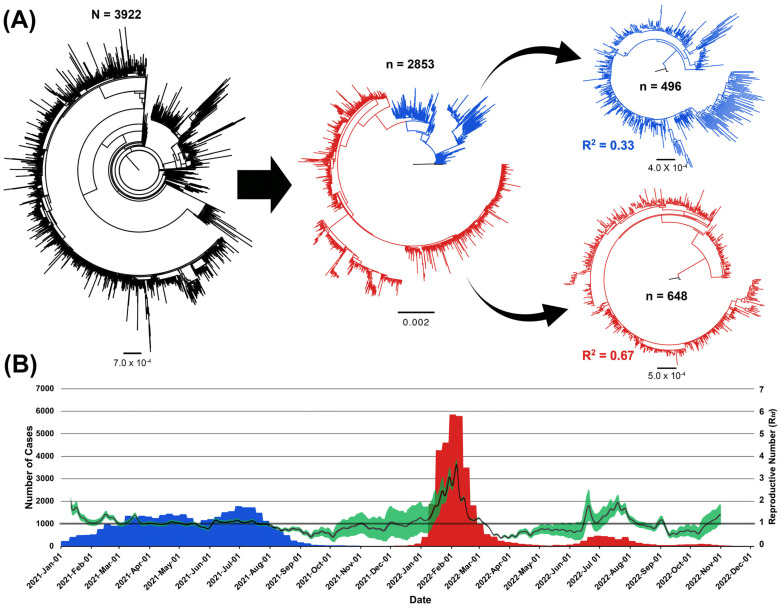
Reconstructed ML trees of Omicron and Delta variants combing local and global sequences datasets and temporal dynamics of SARS-CoV-2 observed cases in Kuwait between January 2021 and November 2022. (**A**) A flow chart of ML trees showing the stages of selecting the best representative collection of sequences and their related lineages for the Omicron and Delta variants isolated in Kuwait and worldwide. The first tree on the top left comprises NextStrain sequences and all Delta and Omicron viruses isolated in Kuwait between December 2019 and October 2022. The ML tree on the middle top is a subsample of NextStrain isolates combined with all Delta and Omicron viruses isolated in Kuwait, which was selected using genome-sampler (G-S) version 2.0. G-S was used again to select the final datasets for the Delta and Omicron variants with their global descendant lineages. Blue branches represent the delta isolates and red branches represent Omicron isolates. All ML trees were reconstructed using the GTR + F + R4 substitution model implemented in IQ-Tree version 2.0. The scale bar below each tree represents the substitution rate per site. Root-to-tip divergence (R2) was estimated using TempEst version 1.5.3. (**B**) Temporal distribution of weekly confirmed SARS-CoV-2 cases between January 2021 and November 2022. Blue bars represent the period when Delta was dominant, while red bars represent the period when Omicron was dominant in Kuwait. The epidemic curve is superimposed by the estimated curve of the time-dependent reproductive numbers (R_td_); the green-shaded areas indicate their 95% confidence interval.

**Figure 2 viruses-16-01872-f002:**
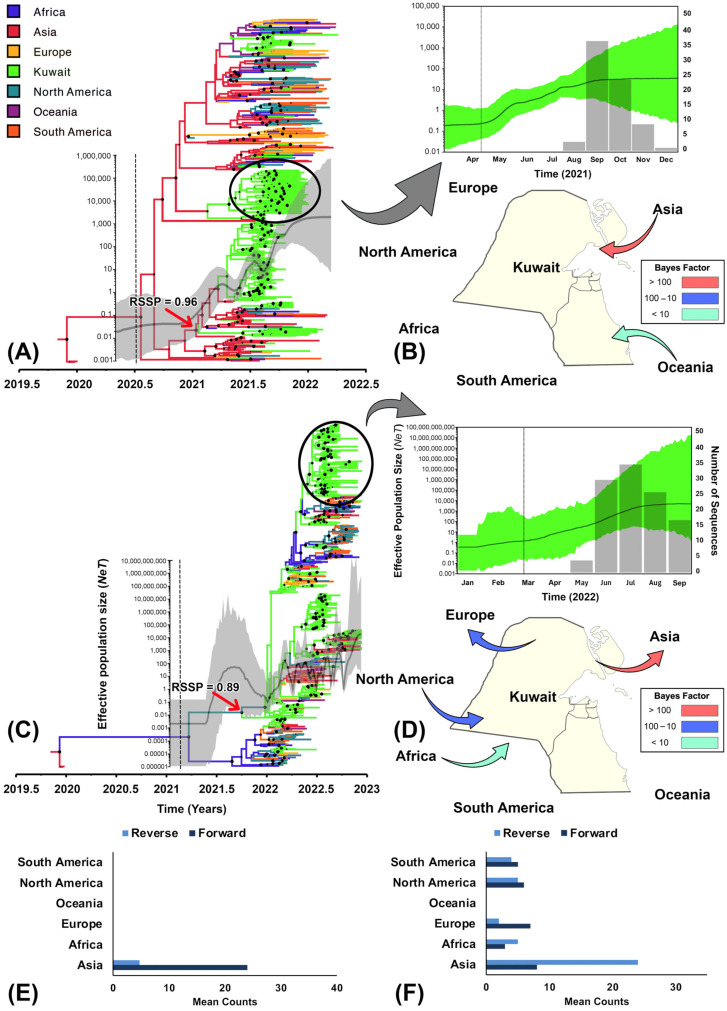
Maximum clade credibility (MCC) phylogeny of the Delta and Omicron variants in Kuwait (with related global isolates), their effective population sizes through time, and regional exchange routes between January 2021 and November 2022. (**A**) represents the MCC tree and Skygrid reconstructions for the Delta variant. (**C**) represents the MCC tree and Skygrid reconstructions for the Omicron variant. The colors of the branches represent the most probable location state of their descendent nodes and correspond to the legend on the upper left. Black dot sizes on the nodes are proportional to the posterior support. The red arrows on the MCC trees point to the earliest probable introduction of each variant to Kuwait from other regions with their inferred root state posterior probability. Green Skygrids plots on the right are reconstructions from the Kuwaiti clades encompassed by black circles. The posterior median estimate is indicated by the dark green line, while the light green shades indicate the 95% highest posterior density. The vertical dotted lines indicate the inferred time each variant transitioned from slow to fast population growth. (**B**,**D**) indicate significant dispersal routes (Bayes factor > 3) of the Delta and Omicron variants, respectively, from and to Kuwait from other regions, and their colors correspond to the magnitude of their significance (legend on the left). (**E**,**F**) are bar charts summarizing the expected reverse and forward Markov jumps for the Delta and Omicron variants, respectively, between Kuwait and other regions.

**Figure 3 viruses-16-01872-f003:**
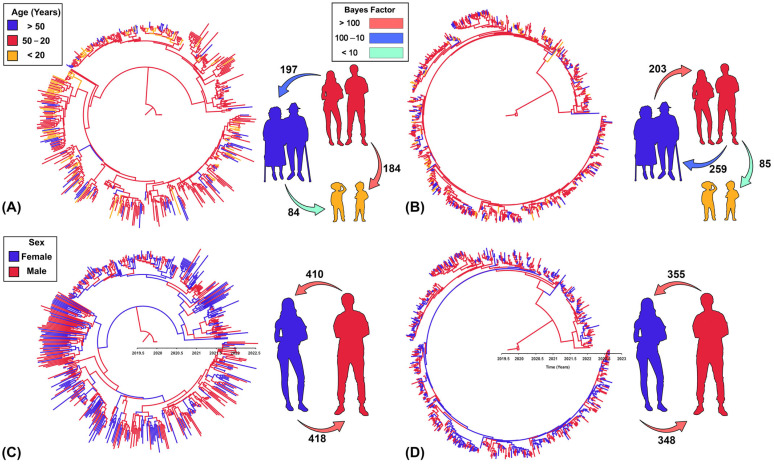
Maximum clade credibility (MCC) trees and host transmission routes over the phylogeny of the Delta and Omicron variants in Kuwait. The colors of the branches indicate the most probable host age (**A**,**B**) and sex (**C**,**D**) state of their descendent nodes and correspond to the legends on the left. (**A**,**C**) represent the delta variant. (**B**,**D**) represent the Omicron variant. Significantly supported transmission (Bayes factors > 10) routes. The numbers adjacent to the arrows indicate the inferred values of the expected reverse and forward Markov jumps between age and sex groups.

**Table 1 viruses-16-01872-t001:** Parsimony scores (Ps) and Association indices (Ai) for the discrete trait models of the Delta and Omicron variants.

Trait		P*s*	95% CI	*p*-Value	A*i*	95% CI	*p*-Value
Delta							
Sex	Observed	208.9	(203.0, 214.0)	0.18	33.8	(31.9, 35.6)	0.19
	Null	205.1	(205.1, 222.0)		35.0	(32.7, 37.2)	
Age	Observed	218.7	(214.0, 223.0)	0.03 *	37.4	(35.7, 39.0)	0.02 *
	Null	226.2	(224.2, 229.0)		41.4	(40.6, 44.9)	
Region	Observed	199.6	(196.0, 203.0)	<0.01 *	33.19	(31.6, 34.8)	<0.01 *
	Null	227.9	(223.1, 232.1)		39.0	(36.9, 40.7)	
Omicron							
Sex	Observed	244.6	(237.0, 252.0)	0.22	38.1	(35.8, 40.5)	0.12
	Null	249.0	(239.2, 256.6)		39.8	(37.5, 42.3)	
Age	Observed	243.6	(238.0, 249.0)	<0.01 *	37.9	(35.6, 40.2)	<0.01 *
	Null	270.2	(265.2, 275.0)		44.4	(42.6, 45.9)	
Region	Observed	217.5	(213.0, 222.0)	<0.01 *	31.3	(29.2, 33.3)	<0.01 *
	Null	253.9	(250.2, 257.3)		43.8	(41.9, 45.7)	

* Statistically significant.

## Data Availability

GISAID Identifier: EPI_SET_240623oy; doi: 10.55876/gis8.240623oy.

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
