# Peer review of "Comparative Evolutionary Epidemiology of SARS-CoV-2 Delta and Omicron Variants in Kuwait"

_viruses, 2024, doi:10.3390/v16121872_

Round 1
Reviewer 1 Report
Comments and Suggestions for Authors
I believe that the manuscript is an interesting addition to the evolutionary pattern of SARS-CoV-2. I personally have no further remarks that would restrain this work from publication.
The manuscript deals with a very important aspect of viral evolution and epidemiology, in my opinion. I do believe that a better understanding of evolutionary epidemiology in either SARS-CoV-2 or any other virus with a significant epidemic impact in our societies is of utmost significance. The specific work is a sound work, written very well in terms of both language and structure (some very minor, rather typographical errors that I am sure will be resolved during the author's proofs).
The variant-specific evolutionary epidemiology of all the Delta and Omicron sequences collected between 2021 and 2023 in Kuwait. I do consider the topic original or relevant to the field. The authors present a very important way to use variant-specific comparative phylodynamic models within local SARS-CoV-2 genomic surveillance efforts in a geographic area. A phylodynamic approach has been used before in viral epidemiology, it is not entirely novel, but it is always important. The authors provide some novel results from a certain geographic area with many important conclusions regarding the devastating COVID epidemic, such as the differences in the evolutionary rates between Delta and Omicron variants and the epidemic routes of transmission.
• No further improvements may be needed, in my opinion.
• The conclusions are consistent with the evidence and arguments presented and the authors address the main question posed. And many more data are being accumulated that support the differential evolutionary rates between the different variants and the significance of such research in the combat against such serious epidemics and pandemics.
• The references are appropriate.
• No further comments on the tables and figures.
Author Response
Reviewer 1
I believe that the manuscript is an interesting addition to the evolutionary pattern of SARS-CoV-2. I personally have no further remarks that would restrain this work from publication.
The manuscript deals with a very important aspect of viral evolution and epidemiology, in my opinion. I do believe that a better understanding of evolutionary epidemiology in either SARS-CoV-2 or any other virus with a significant epidemic impact in our societies is of utmost significance. The specific work is a sound work, written very well in terms of both language and structure (some very minor, rather typographical errors that I am sure will be resolved during the author's proofs).
The variant-specific evolutionary epidemiology of all the Delta and Omicron sequences collected between 2021 and 2023 in Kuwait. I do consider the topic original or relevant to the field. The authors present a very important way to use variant-specific comparative phylodynamic models within local SARS-CoV-2 genomic surveillance efforts in a geographic area. A phylodynamic approach has been used before in viral epidemiology, it is not entirely novel, but it is always important. The authors provide some novel results from a certain geographic area with many important conclusions regarding the devastating COVID epidemic, such as the differences in the evolutionary rates between Delta and Omicron variants and the epidemic routes of transmission.
• No further improvements may be needed, in my opinion.
• The conclusions are consistent with the evidence and arguments presented and the authors address the main question posed. And many more data are being accumulated that support the differential evolutionary rates between the different variants and the significance of such research in the combat against such serious epidemics and pandemics.
• The references are appropriate.
• No further comments on the tables and figures
Authors’ response: We thank the reviewer for their valuable time in reviewing the manuscript and we are glad that they liked it.
Reviewer 2 Report
Comments and Suggestions for Authors
The authors analyze SARS-CoV-2 sequence data from public database to analyze epidemiology of SARS-CoV-2 in Kuwait. Phylodynamic approaches was utilized to combine evolutionary, demographic and epidemiological ideas to obtain valuable information about the evolution and transmission of SARS-CoV-2 within well-defined geographic area. The authors successfully presented what kind of information is available to obtain from genomic data accumulated during the pandemic when phylodynamic analysis is applied with sophisticated statistical methodology. Although there is no novel findings reported in this manuscript, this reports will provide basis for further investigation.
Minor points.
1. Line 294, (B) > (C)
2. Line 302; Please add indication that (B) is for Delta (C) for is Omicron.
3. Line 304; Please add indication that (E) is for Delta (F) for is Omicron.
4. Line 317; Is April 2021 correct?
Author Response
Reviewer 2
The authors analyze SARS-CoV-2 sequence data from public database to analyze epidemiology of SARS-CoV-2 in Kuwait. Phylodynamic approaches was utilized to combine evolutionary, demographic and epidemiological ideas to obtain valuable information about the evolution and transmission of SARS-CoV-2 within well-defined geographic area. The authors successfully presented what kind of information is available to obtain from genomic data accumulated during the pandemic when phylodynamic analysis is applied with sophisticated statistical methodology. Although there is no novel findings reported in this manuscript, this reports will provide basis for further investigation.
Authors’ response: We thank the reviewer for their valuable time in reviewing the manuscript and we are glad that they liked it.
Minor points.
- Line 294, (B) > (C)
Line 294: The figure’s caption has been corrected as indicated by the reviewer.
- Line 302; Please add an indication that (B) is for Delta and (C) for Omicron.
Line 302: The figure’s caption has been corrected as indicated by the reviewer.
- Line 304; Please add indication that (E) is for Delta (F) for is Omicron.
Line 304: The figure’s caption has been corrected as indicated by the reviewer.
- Line 317; Is April 2021 correct?
Line 317: Yes, its correct, but maybe the sentence was confusing. Thus, we fixed the corresponding sentence to clearly reflect the Skygrid analysis results in Figure 2B.